# Characterization of CRISPR-Cas Systems in *Serratia marcescens* Isolated from *Rhynchophorus ferrugineus* (Olivier, 1790) (Coleoptera: Curculionidae)

**DOI:** 10.3390/microorganisms7090368

**Published:** 2019-09-19

**Authors:** Maria Scrascia, Pietro D’Addabbo, Roberta Roberto, Francesco Porcelli, Marta Oliva, Carla Calia, Anna Maria Dionisi, Carlo Pazzani

**Affiliations:** 1Department of Biology, University of Bari Aldo Moro, 70124 Bari, Italy; maria.scrascia@uniba.it (M.S.); pietro.daddabbo@uniba.it (P.D.); marta.oliva@uniba.it (M.O.); carla.calia@virgilio.it (C.C.); 2Department of Plants, Food, and Soil Sciences, University of Bari Aldo Moro, 70124 Bari, Italy; roberta.roberto@uniba.it (R.R.); francesco.porcelli@uniba.it (F.P.); 3Department of Infectious diseases, Istituto Superiore di Sanità, 00161 Rome, Italy; annamaria.dionisi@iss.it

**Keywords:** CRISPR-Cas systems, subtype I-E, subtype I-F, *Serratia marcescens*, symbiont, *Rhynchophorus ferrugineus*

## Abstract

The CRISPR-Cas adaptive immune system has been attracting increasing scientific interest for biological functions and biotechnological applications. Data on the *Serratia marcescens* system are scarce. Here, we report a comprehensive characterisation of CRISPR-Cas systems identified in *S. marcescens* strains isolated as secondary symbionts of *Rhynchophorus ferrugineus*, also known as Red Palm Weevil (RPW), one of the most invasive pests of major cultivated palms. Whole genome sequencing was performed on four strains (S1, S5, S8, and S13), which were isolated from the reproductive apparatus of RPWs. Subtypes I-F and I-E were harboured by S5 and S8, respectively. No CRISPR-Cas system was detected in S1 or S13. Two CRISPR arrays (4 and 51 spacers) were detected in S5 and three arrays (11, 31, and 30 spacers) were detected in S8. The CRISPR-Cas systems were located in the genomic region spanning from *ybhR* to *phnP*, as if this were the only region where CRISPR-Cas loci were acquired. This was confirmed by analyzing the *S. marcescens* complete genomes available in the NCBI database. This region defines a genomic hotspot for horizontally acquired genes and/or CRISPR-Cas systems. This study also supplies the first identification of subtype I-E in *S. marcescens*.

## 1. Introduction

Prokaryotic genomic loci, named Clustered Regularly Interspaced Short Palindromic Repeats (CRISPR)-CRISPR associated proteins (CRISPR-Cas system) are adaptive immune systems that confer resistance to the entry of intracellular parasitic elements like phages and plasmids [1,2,3]. These systems might also play additional functions, such as regulation of virulence and the stress response [4,5,6,7]. The CRISPR-Cas systems consist of at least one array where DNA sequences of 25–35 bp each (direct repeats-DRs) with a high level of nucleotide identity are regularly interspaced by unique sequences, termed spacers (typically 30–40 bp each), that may differ considerably, depending on their extracellular origin [8].

The CRISPR arrays are generally flanked at one end by an AT-rich, non-coding region (leader sequence) extending from 47 bp to a few hundred bp, which promotes CRISPR transcription and guides the insertion of new spacers [9,10,11,12]. Single or multiple CRISPR arrays may be found adjacent to Cas proteins; the latter mediate adaptive immunity and various connected functions [7,13,14]. The Cas proteins are grouped into two functional modules [14,15,16]: (i) An adaptation module that delivers exogenous DNA sequences into CRISPR arrays, and (ii) effector modules that target and cleave invading nucleic acids. The adaptation module, mainly composed of Cas1 and Cas2 proteins, is conserved, while the effector modules are highly variable among the CRISPR-Cas systems described in the literature.

Based on the different gene organizations of the effector modules, the CRISPR-Cas systems are currently grouped into classes, types, and subtypes. Class 1 includes three types (I, III, and IV) which, in turn, are subdivided into eight, five, and two subtypes, respectively. Class 2 is composed of three types (II, V, and VI) subdivided into four, ten, and four subtypes, respectively [14,15,16]. CRISPR-Cas systems are widespread in prokaryotes; the Enterobacteriaceae bacterial family has recently been reviewed by Medina-Aparicio et al., [17]. These systems were detected in 57% of the analyzed genomes, with subtypes I-E and I-F being almost exclusively identified. I-E was mainly detected in *Escherichia*, *Klebsiella*, *Salmonella,* and *Shigella* genera and I-F was detected in *Yersinia* and, to a lesser extent, in *Escherichia* genera. However, for the genus *Serratia* and its type species *marcescens*, data on CRISPR-Cas systems are still very scarce. Indeed, the only descriptions of CRISPR-Cas systems in *S. marcescens* are those reported by: (1) Vicente et al., 2016 [18] on the presence of a CRISPR-Cas system subtype I-F in the *S. marcescens* strain PWN146, isolated from the nematode *Bursaphelenchus xylophilus*; (2) Srinivasan et al., 2018 [19] who reported the presence of *cas1*, *cas3,* and additional *csy* genes (*csy*1–4) in the *S. marcescens* strain SM03 isolated from human gut; and (3) Medina-Aparicio et al., 2018 2018 [17] who mentioned the presence of the CRISPR-Cas system subtype I-F in the *S. marcescens* strain FG194.

The aims of this study were: (i) to identify and characterize the CRISPR-Cas systems in *S. marcescens* symbiont strains of *Rhynchophorus ferrugineus*, also known as Red Palm Weevil (RPW), one of the most invasive pests of major cultivated palms; (ii) to identify the genomic region(s) of *S. marcescens* involved in horizontal acquisition of CRISPR-Cas systems; and iii) to provide data on the symbiotic relationship between RPW and *S. marcescens* (e.g., the clonal relationship of bacteria symbionts).

## 2. Materials and Methods

### 2.1. Serratia marcescens Strains

*S. marcescens* was regularly isolated from the reproductive apparatus of adult RPWs, deposited eggs, and along the tissue of infested palms. Isolation was performed in Southern Italy from 2009 to 2014 [20]. Based on different places and years of isolation (Table 1), 4 strains (S1, S5, S8, and S13), collected from the reproductive apparatus of RPWs were randomly selected for whole genome sequencing and detection of CRISPR-Cas systems.

### 2.2. Genome Sequencing and Characterization of CRISPR-Cas Systems

Genomic DNA was purified using the cetyltrimethylammonium bromide method [21] and sequenced using the SELGE (“Laboratory network for the selection, characterization and conservation of germplasm and for preventing the spread of economically-relevant and quarantine pests” No. 14, subsidised by the Apulia Region, PO FESR 2007–2013—Axis I, Line of intervention 1.2, Action 1.2.1) using a 100 bp paired end protocol on an Illumina HiScanSQ platform. Whole genome sequencing reads of each strain were assembled in multiple contigs using the MyPro pipeline [22], failing to create chromosome-wide contigs due to the presence of repeated sequences (e.g., IS). Contigs positive for CRISPR-Cas systems were annotated using Prokka [23] and were graphically displayed using SnapGene software, version 4.3.11 (GSL Biotech LLC, Chicago, IL; https://www.snapgene.com/). The sequence of the S5 contig where the CRISPR-Cas system was identified was found to be interrupted in CRISPR array S5.2 and completed by primer walking of the ~3.2 kb amplicon from nucleotide 30,974 to 34,322. PCRs were performed in a total volume of 50 μL containing 50–100 ng of genomic DNA of strain S5 using 1X PCR buffer (20 mM Tris–HCl, pH 8.4, and 50 mM KCl), 1.5 mM MgCl_2_, 125 μM of deoxynucleotide triphosphate mix (EuroClone S.p.A., Italy), 20 μM of each primer and 1U Platinum *Taq* DNA polymerase (Invitrogen, Thermo Fisher Scientific, Italy). Cycling conditions were: 94 °C for 5 min; 5 cycles of 94 °C for 30 s, 65 °C for 10 s, and 72 °C for 4 min; 10 cycles of 94 °C for 30 s, 65 °C decreasing to 60 °C (0.5 °C/cycle) for 10 s, and 72 °C for 4 min; and 15 cycles of 94 °C for 30 s, 60 °C for 10 s, and 72 °C for 4 min with final extension at 72 °C for 5 min. Primer walking was performed by GENEWIZ European Headquarters (UK). The primers used in this study are reported in Appendix A. The sequence of S1, S5, S8, and S13 contigs harbouring CRISPR-Cas systems and spanning from *tRNA*-*Leu* to *phnF* was submitted to NCBI under the accession numbers MK507743, MK507745, MK507744, and MK507746, respectively. 

CRISPR arrays and *cas* genes were identified by Prokka and three online suites were used to refine the CRISPR arrays annotation and *cas* detection: CRISPRSuite (http://bioanalysis.otago.ac.nz/CRISPRTarget/CRISPRSuite.html) [24], CRISPRone (http://omics.informatics.indiana.edu/CRISPRone/) and CRISPRFinder (http://crispr.i2bc.paris-saclay.fr/) [25]. The names of the *cas* genes are consistent with the classification proposed by Makarova et al. [14,15].

Bioinformatic analysis of the spacer sequences was performed by CRISPRTarget in GenBank-PHAGE and RefSeq-PLASMID databases [26]. In order to establish if the spacers’ sequences were shared among array(s), a pairwise comparison was performed by NCBI Blast. Two spacers with at least 95% identity and less than 10% difference in length were considered to be the same [27]. The online tool WebLogo (http://weblogo.berkeley.edu/logo.cgi) was used to analyze the conservation of DRs of CRISPR arrays [28].

AT rich, non-coding sequences placed upstream and downstream of CRISPR arrays were selected to detect putative leader sequences. The AT% was calculated and compared with the AT% of the whole-contig sequence. Putative leader sequences were screened for prokaryotic promoters using the BDGP Neural Network Promotor Prediction tool (http://www.fruitfly.org/seq_tools/promoter.html).

The phylogenetic tree was produced using BLAST pairwise alignments, and the Blast Tree View Widget. BLAST computes a pairwise alignment between a query (e.g., S1 contig) and a dataset of sequences (i.e., the other selected regions in this study). The algorithm used to produce the phylogenetic tree involved setting the Fast-Minimum Evolution [29] to 0.75, the maximum allowed fraction of mismatched bases in the aligned region between any pair of sequences.

### 2.3. PFGE

Genomic restriction was performed according to the standardised PulseNet *Salmonella* protocol [30]. Agarose-embedded DNA was digested with 40 U of *SpeI* for 3 h at 37 °C. The restriction fragments were separated by electrophoresis in Tris-borate-EDTA (44.5 mM Tris-borate, 1 mM EDTA; pH 8.0) at 14 °C using a CHEF-DRIII (Bio-Rad, Milan, Italy). Electrophoresis conditions were as follows: (Block I) initial switch time 2 s, final switch time 10 s, voltage 6 V, included angle 120°, and run time 13 h; (block II) initial switch time 20 s, final switch time 25 s, voltage 6 V, included angle 120°, and run time 6 h. The size standard used for all gels was *XbaI*-digested DNA from *Salmonella* Braenderup strain, the universal size standard used by all PulseNet laboratories. The PFGE agarose gels were stained with ethidium bromide (40 µg/mL) and the DNA band images were acquired by the Gel Doc-It photo documentation system (Gel Doc-It photo documentation system, UVP, Upland, CA, USA). A dendrogram of pulsotype relationships was developed through the unweighted pair group method using arithmetic averages (UPGMA) with BioNumerics software version 6.5 (Applied Maths). Pulsotypes were assigned to the same clusters if they exhibited 80% similarity in the dendrogram. 

## 3. Results

### 3.1. CRISPR-Cas Systems in S. marcescens from RPW

CRISPR-Cas systems were detected in two (S5 and S8) of the four strains whose genomes were sequenced in this study. The systems were chromosomally located within a variable region that extended from *ybhR* (encoding for an inner membrane transport permease) to *phnP* (encoding for a protein involved in the phosphonate metabolism). Outside of this region (i.e., from the gene for tRNA^Leu^ to *phnF*) the gene organization was found conserved. A detailed description of this region, the *cas* genes organization, and their positions on the corresponding contig are reported in Figure 1 and Appendix A.

S5 harboured a CRISPR-Cas system subtype I-F and had two arrays (S5.1 and S5.2) containing 4 and 51 DRs, respectively. Consensus direct repeats (CDRs) of 28 bp were identified for both arrays (Appendix A). The I-F *cas* operon (*cas1-cas3-cas8f-cas5-cas7-cas6f*) was found between the arrays S5.1 and S5.2 and a putative leader sequence of 131 bp was detected upstream of each array (Appendix A). Of the 53 identified spacers, 19 showed putative origins from phages (6), plasmids (11), or both (2) (Table 2).

S8 harbored the CRISPR-Cas system subtype I-E. Three arrays named S8.1, S8.2, and S8.3 contained 31, 11, and 30 DRs, respectively. CDRs of 29 bp were identified for the three arrays (Appendix A). The subtype I-E *cas* operon (*cas3-cas8e-cse2-cas7-cas5-cas6-cas1-cas2*) was found between S8.1 and S8.2 with an 84 bp putative leader sequence detected upstream of each array (Appendix A). Of the 69 identified spacers, 27 showed putative origins from phages (10), plasmids (13), or both (4) (Table 2). All of the spacers were different. Most of them matched DNA sequences of plasmids mainly isolated from bacteria of the Enterobacteriaceae family. Likewise, phage genomes recognised by spacers came from phages predominantly detected in bacteria of the Enterobacteriaceae family and, to a lesser extent, from other bacteria families (e.g., Burkholderiaceae, Pseudomonadaceae, and Vibrionaceae) (Appendix A). Nine spacers hit genetic elements identified in *S. marcescens* and seven matched plasmids, of which one (pSMC2) was conjugative [31] and one (pCAV1492-199) was reported to harbour copper resistance genes [32]. Two spacers hit phages, of which one (phage Eta) might establish unstable lysogeny with the host [33]. Interestingly, phage Eta genome was recognised by six spacers harboured by S8 arrays.

### 3.2. CRISPR-Cas Systems in S. marcescens

A total of 15 *S. marcescens* complete genomes available in the NCBI database (www.ncbi.nlm.nih.gov/genome/genomes/1112; version Dec. 2018) were investigated to detect the presence of CRISPR-Cas systems or orphan CRISPR arrays (i.e., arrays without adjacent *cas* genes). This analysis, to the best of our knowledge, has never before been performed. Eight strains were found positive (Table 3).

Three strains, two environmental (PWN146 and N4-5) and one clinical (12TM), harboured a CRISPR-Cas subtype I-F. Five isolates harboured orphan CRISPR arrays. The I-E subtype was detected in two environmental (KS10 and EL1) and two clinical (CAV1492 and CAV1761) strains and the I-F subtype was found in the environmental strain B3R3. It is noteworthy that I-E was only identified in orphan CRISPR arrays. Conservation of nucleotide positions within the DRs identified in this study was analyzed by WebLogo and reported in Appendix A. Subtypes I-E and I-F were localised within the same *ybhR-phnP* chromosomal region already identified for S5 and S8 strains. We analyzed this region present in complete genomes of *S. marcescens* strains devoid of detectable CRISPR-Cas systems or orphan CRISPR arrays. Apart from the presence of different CRISPR-Cas systems or orphan CRISPR arrays, this region was found to harbour genes present in specific strains (Appendix A). Taken together, these data suggest that the chromosomal region between *ybhR* and *phnP* is a hotospot for horizontally acquired genes and/or CRISPR-Cas systems.

The region from the gene for tRNA^Leu^ to *phnF* was selected to establish the clonal relationship between our strains and those of *S. marcescens* that we took for comparison. S1, S5, S8, and S13 were distributed along four distant branches (Figure 2). The clonal relationship was also established by analysis of S1, S5, S8, and S13 PFGE profiles, which assigned these strains to different clusters (% of similarity lower than 80) (Appendix A). A detailed description of the region from gene for tRNA^Leu^ to *phnF* is reported in Appendix A.

## 4. Discussion

*Serratia* is a ubiquitous genus in nature. *Serratia* species have been isolated from water, soil, animals (including people), and from the surface of plants [34]. In insects, *Serratia* spp. have been reported either as pathogens or symbionts, with *marcescens* and *liquefaciens* being the species predominantly identified [34]. For instance, *S. liquefaciens* and *S. marcescens* were found in sugar-beet root-maggot development stages (*Tetanops myopaeformis*), suggesting an insect microbe symbiosis as well as a nutritional interdependence [35]. *S. marcescens* was been found to be facultatively associated with the plant-nematode *Bursaphelenchus xylophilus* [18]. Concerning this, we recently reported *S. marcescens* as a secondary symbiont of the RPW [20], one of the most invasive pests of major cultivated palms.

Data of CRISPR-Cas systems in *S. marcescens* are still scarce. Despite being limited, the analysis of the four *S. marcescens* genome sequences considered in this study shows the presence of either subtype I-E or subtype I-F. Detection of subtype I-E, to the best of our knowledge, has not previously been reported in this species. *In silico* analysis revealed the unicity of all the spacers and memories of different foreign genetic elements that invaded S5 and S8.

The subtypes I-E and I-F identified in our strains and in the available complete genomes of *S. marcescens* were located within the genomic region spanning from *ybhR* to *phnP*. This region is likely to be a genomic hotspot for horizontally acquired genes and/or CRISPR-Cas systems. The tendency of CRISPRs to be distributed non-randomly on bacterial chromosomes, but occurring in specific chromosome regions, has already been reported [36,37,38]. For instance, within the Enterobacteriaceae family genome, hotspots involved in the acquisition of CRISPR-Cas systems have been described for *Klebsiella* spp. [39,40] and *Escherichia coli* [41]. Our findings extend this knowledge to *S. marcescens*.

We also performed a phylogenetic analysis by using the genomic sequence encompassing *ybhR*-*phnP* and extending from the gene for tRNA^Leu^ to *phnF*. In the resulting dendrogram, our strains were situated on distant branches. The clonal relationship was further established by analysis of the S1, S5, S8, and S13 pulsotypes that assigned these bacteria to different clonal groups. The unrelated clonal relationship, the different immune memories, and the presence of different CRISPR subtypes or their absence suggest that RPW might have acquired *S. marcescens* through independent events. However more data are necessary to support this hypothesis.

CRISPR studies are receiving a lot of interest from the scientific community, particularly for their potential applications in different fields [42,43]. However, studies of CRISPR diversity can also greatly help in the understanding of evolutionary dynamics of specific bacteria populations [44]. The symbiotic association between eukaryotes and prokaryotes is now recognised as one of the main forces shaping life on our planet [45]. Concerning this, the data presented here supply both a comprehensive characterization of CRISPR-Cas systems in *S. marcescens* symbionts of RPW and the first comparative analysis of CRISPR loci in *S. marcescens*.

## Figures and Tables

**Figure 1 microorganisms-07-00368-f001:**
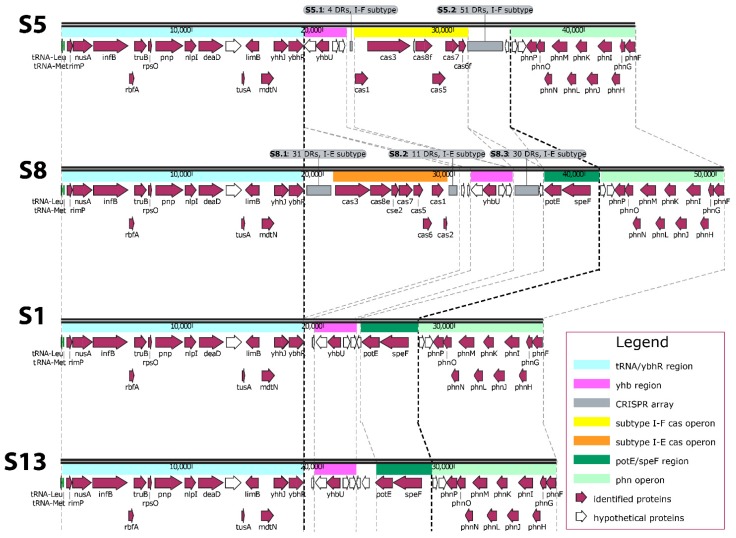
Comparative analysis of the genomic region extending from the gene for tRNA^Leu^ to *phnF* in S1, S5, S8, and S13. The region was labelled both by automatic Prokka annotation and manual editing. Dashed lines follow the corresponding region among strains. Bold dashed lines delimit the variable domain. The array code, number of direct repeats, and CRISPR subtype are reported in the grey flags on the top of the corresponding CRISPR array. Genes and open reading frames are represented by arrow boxes pointing in the direction of transcription. Feature names are labelled below the arrow boxes.

**Figure 2 microorganisms-07-00368-f002:**
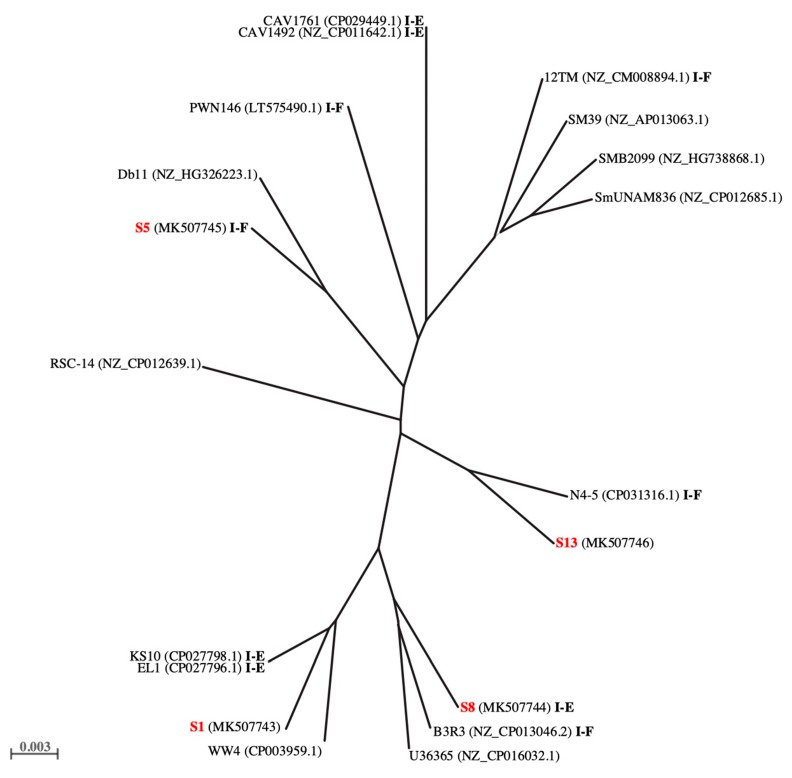
Phylogenetic tree. The phylogenetic tree was obtained from comparative analysis of the genomic region extending from the gene for tRNA^Leu^ to *phnF* in S1, S5, S8, and S13 (in red) and *S. marcescens* complete genomes available from the NCBI database. System types are highlighted in bold.

**Table 1 microorganisms-07-00368-t001:** *Serratia marcescens* strains analyzed in this study.

Strain	Country	Place (Region)	Year	Source
S1	Italy	Valenzano (Apulia)	2009	Female egg-chamber
S5	Italy	Bari (Apulia)	2014	Male reproductive apparatus
S8	Italy	Salerno (Campania)	2013	Female ovipositor
S13	Italy	Bari (Apulia)	2014	Female copulatory pouch

**Table 2 microorganisms-07-00368-t002:** Origin of spacers found in CRISPR-Cas systems of *S. marcescens* strains S5 and S8.

Array	Origin and Spacer Index		
	Phage	Plasmid	Phage and Plasmid
S5.1	none	2, 3	none
S5.2	1, 2, 22, 23, 41, 48	13, 15, 26, 29, 32, 35, 36, 43, 49	20, 47
S8.1	2, 18, 19	3, 6, 8, 9, 12, 13, 16, 17, 20, 23, 30	5, 29
S8.2	none	8, 10	9
S8.3	1, 2, 3, 5, 15, 17, 19	none	4

**Table 3 microorganisms-07-00368-t003:** CRISPR subtypes identified in complete genomes of *S. marcescens* available in the NCBI database.

Strain	Source	Accession Number	Subtype
12TM	clinical	NZ_CM008894.1	I-F
B3R3	environmental	NZ_CP013046.2	I-F
CAV1492	clinical	NZ_CP011642.1	I-E
CAV1761	clinical	CP029449.1	I-E
EL1	environmental	NZ_CP027796.1	I-E
KS10	environmental	NZ_CP027798.1	I-E
N4-5	environmental	NZ_CP031316.1	I-F
PWN146	environmental	LT575490.1	I-F

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
