# Peer review of "Characterization of CRISPR-Cas Systems in Serratia marcescens Isolated from Rhynchophorus ferrugineus (Olivier, 1790) (Coleoptera: Curculionidae)"

_microorganisms, 2019, doi:10.3390/microorganisms7090368_

Round 1

Reviewer 1 Report

Overall, the authors report a largely descriptive account of sequencing four new isolates of Serratia marcescens and of the identification of CRISPR-Cas systems within these strains. Other than repeated claims of being the ‘first to look for CRISPR-Cas systems in S. marcescens’, the rationale for performing the study is unclear. What is the bigger-picture gap in knowledge that the authors hope to addresses? How does the reporting of these systems advance understanding of how the systems function, how they are acquired, or their biological role(s) in this particular system? Perhaps the most interesting finding is a genomic ‘hotspot’ for the acquisition of horizontally acquired genes, such as CRISPR-Cas systems, as has been reported in many other studies of horizontal gene acquisition in divese bacteria. However, this finding in S. marcescens is only superficially examined.

Major comments:

Figure 2: What does this tell us that isn’t already communicated in Fig1?

Figure 3: What is the key message here, why is this interesting/what does it tell us? Probably not worth a full-page figure. Inclusion of comparison with CRISPRs found in the additional genomes in Figure 4 might be informative.

Figure 4: It is unclear what these data are based on. Is this the ANI between the genomic region indicated? Have you looked at bootstrapping the tree? How does the phylogeny of the acquired region compare with phylogeny of the rest of the strains’ genomes? What is the point of presenting this tree? Perhaps overlaying the systems types found would be informative.

It is also unclear why the authors performed PFGE of the strains when they had complete genomes sequences available. Why not perform an ANI analysis or similar?

Discussion: There appears to be very little attempt to consider what the data presented tell us about the underlying biology at play. For example, L269 mentions in silico analyses of spacer targets. Presumably, this refers to the supplementary data of the CRISPRTarget output. However, no attempt is made to dig into what phages the spacers match, e.g. are there any known Serratia phages? Are any of the spacers shared between strains? The lack of advancement is highlighted by the concluding sentence of the Discussion: “…. thus providing evolutionary inputs on the RPW-S. marcescens relationship”. What does this sentence mean?

Minor editorial comments:

L71: remove ‘simply’

L72 cys -> csy

L78 – 81: This sentence is unclear, pleas repharase. Also, the ‘first time’ claim should be removed.

L108 (and elsewhere): 1,5 mM -> 1.5 mM and µmol/L -> µM

L120: Add citation for CRISPRDectect

L125: Add citation for CRISPRTarget

L176: cas3 -> cas2-3 (fusion in type I-F systems)

L219: Remove this sentence.

L256: man -> people.

Author Response

REVIEWER 1

COMMENTS AND SUGGESTIONS FOR AUTHORS

Comment 1: …the rationale for performing the study is unclear. What is the bigger-picture gap in knowledge that the authors hope to address?

Reply: description and characterisation of CRISPR-Cas systems in S. marcescens is very scarce (as highlighted in the introduction). The rationale of this study was to provide data that can contribute to better describe the CRISPR-Cas systems (e.g. their organization) present in this species. Data will also supply knowledge on the genome region involved in the acquisition of CRISPR-Cas systems and on the spacers’ origin. This has now been better specified in the sections introduction, results and discussion.

Comment 2: How does the reporting of these systems advance understanding of how the systems function, how they are acquired, or their biological role(s) in this particular system?

Reply: in this study we did not intend to investigate how the CRISPR-Cas systems identified in S. marcescens symbiont strains of RPW function or how they were acquired.

Comment 3: Perhaps the most interesting finding is a genomic ‘hotspot’ for the acquisition of horizontally acquired genes, such as CRISPR-Cas systems, as has been reported in many other studies of horizontal gene acquisition in diverse bacteria. However, this finding in S. marcescens is only superficially examined.

Reply: we thank the reviewer for this valuable suggestion. The identified genomic hotspot for horizontal acquisition of CRISPR-Cas systems, as well as additional genes, has now been better highlighted in the manuscript (sections abstract, introduction, results and discussion).

MAJOR COMMENTS

Figure 2: What does this tell us that isn’t already communicated in Fig1?

Reply: Figure 2 has been deleted

Figure 3

Comment 4: What is the key message here, why is this interesting/what does it tell us? Probably not worth a full-page figure. Inclusion of comparison with CRISPRs found in the additional genomes in Figure 4 might be informative.

Reply: in Figure 3 we showed the nucleotide conservation of each base within the DRs of arrays. This figure has now been removed from the main text and presented in supplementary data as Figures S1 (subtype I-E) and S2 (subtype I-F). In these figures we also report (as requested) the WebLogo of DRs identified in the complete genomes of S. marcescens available in the NCBI database.

Figure 4

Comment 5: It is unclear what these data are based on. Is this the ANI between the genomic region indicated? Have you looked at bootstrapping the tree?

Reply: Figure 4 has now been renamed as Figure 2. The tree was produced using BLAST pairwise alignments, and the Blast Tree View Widget. BLAST computes a pairwise alignment between a query (e.g. S1 contig) and a dataset of sequences (i.e. the other selected regions in this study). The algorithm used to produce the tree was the Fast-Minimum Evolution (Desper and Gascuel, 2004; DOI: 10.1093/molbev/msh049) setting to 0.75 the maximum allowed fraction of mismatched bases in the aligned region between any pair of sequences. This sentence was added in the section materials and methods. The sentence “The genetic relationship was analysed by the NCBI Tree Viewer (Treeviewer JS version:1.17.4) and manually revised” was removed from the legend of Figure 2.

Comment 6: How does the phylogeny of the acquired region compare with phylogeny of the rest of the strains’ genomes? What is the point of presenting this tree?

Reply: S. marcescens has been reported as a secondary symbiont of RPW. Acquisition of this symbiont might have occurred either by single or independent events. Our hypothesis is that acquisition occurred by independent events. We based this assumption on the different clonal relationship of our strains established by: analysis of the PFGE profiles; distribution of strains on distant branches (as highlighted by the phylogenetic tree presented in Figure 2); the presence of different CRISPR-Cas subtypes or their absence.

Comment 7: Perhaps overlaying the systems types found would be informative.

Reply: the CRISPR system types are now present in Figure 2.

Comment 8: It is also unclear why the authors performed PFGE of the strains when they had complete genomes sequences available. Why not perform an ANI analysis or similar?

Reply: genome sequences of our four strains were not fully assembled as chromosome-wide contig. This has now been specified in materials and methods. In this study analysis of PFGE pulsotypes allows to establish the unrelated clonal relationship among S. marcescens symbiont strains of RPW regardless of fully assembled chromosome-wide contigs availability. Since the clonal relationship established by analysis of the PFGE profiles and comparison of the region spanning from tRNALeu to phnF (phylogenetic tree presented in Figure 2) was proved to be unrelated, we believe these data should be sufficient for aim iii) of this study.

Discussion

Comment 9: There appears to be very little attempt to consider what the data presented tell us about the underlying biology at play. For example, L269 mentions in silico analyses of spacer targets. Presumably, this refers to the supplementary data of the CRISPRTarget output. However, no attempt is made to dig into what phages the spacers match, e.g. are there any known Serratia phages? Are any of the spacers shared between strains?

Reply: we thank the reviewer for this valuable suggestion. List of the plasmids and/or phages the spacers match is reported in File S2. Their distribution, between arrays, is now added in the same file. A description of spacers matching plasmids and phages detected in S. marcescens is now reported in the section results. All the spacers are different, according to the criteria reported by Touchon et al., 2011 (doi:10.1128/JB.01307-10). This has now been specified in the sections materials and methods and results.

Comment 10: The lack of advancement is highlighted by the concluding sentence of the Discussion: “…. thus providing evolutionary inputs on the RPW-S. marcescens relationship”. What does this sentence mean?

Reply: the sentence has been removed

MINOR EDITORIAL COMMENTS:

L71: remove ‘simply’

Reply: ‘simply’ was removed

L72 cys -> csy

Reply: cys was corrected with csy

L78 – 81: This sentence is unclear, pleas repharase. Also, the ‘first time’ claim should be removed.

Reply: the sentence was completely rephrased

L108 (and elsewhere): 1,5 mM -> 1.5 mM and µmol/L -> µM

Reply: corrected

L120: Add citation for CRISPRDetect

Reply: citation was added

L125: Add citation for CRISPRTarget

Reply: citation was added

L176: cas3 -> cas2-3 (fusion in type I-F systems)

Reply: cas3 was corrected with cas2-3

L219: Remove this sentence.

Reply: the sentence was removed

L256: man -> people.

Reply: man was replaced with people

Reviewer 2 Report

Scrascia et al report sequencing and characterization of CRISPR-Cas systems from two S. marcescens strains. The task is conceptually simple, adequately performed and presented with appropriate clarity. I have only a few minor comments.

- In the abstract the authors claime to have done genome sequencing of four S. marcescens strains. Further in the text (ll. 115-117), it turns out that only relatively short fragments, relevant to the CRISPR-Cas locus, are deposited to GenBank. Did the authors sequenced the whole genomes, and if yes, are there any particular reasons why the whole sequences were not deposited? If not, maybe it would be best to say something like "partial chromosome sequencing" in the abstract.

- According to Figure 3, the CRISPR repeat of the S5.1 array is considerably less conserved compared to those of other arrays in these S. marcescens strains, as well as compared to what's typical for CRISPR repeats in general. In conjunction with the lack of obvious phage-derived spacers (Table 2) and with its length, how sure are the authors that this is a real array? Any additional evidence would be welcome.

- The tree in Figure 4 is shown as rooted, with S. marcescens RSC-14 as an outgroup. Was there any particular reason to place the root there? If the tree is actually unrooted, it should rather be depicted as such.

Author Response

REVIEW 2

COMMENTS AND SUGGESTIONS FOR AUTHORS

Comment 1: In the abstract the authors claime to have done genome sequencing of four S. marcescens strains. Further in the text (ll. 115-117), it turns out that only relatively short fragments, relevant to the CRISPR-Cas locus, are deposited to GenBank. Did the authors sequence the whole genomes, and if yes, are there any particular reasons why the whole sequences were not deposited?

Reply: WGS reads of each strain were assembled in multiple contigs using the MyPro pipeline, failing to create chromosome-wide contigs due to the presence of repeated sequences (e.g. IS). This has now been specified in the section materials and methods. Considering the aims of the study, only the contigs containing CRISPR-Cas systems were deposited.

L115-117: the sentence was amended

Comment 2: ….maybe it would be best to say something like "partial chromosome sequencing" in the abstract.

Reply: "Whole genome sequencing" was added in the abstract.

Comment 3: According to Figure 3, the CRISPR repeat of the S5.1 array is considerably less conserved compared to those of other arrays in these S. marcescens strains, as well as compared to what's typical for CRISPR repeats in general.

Reply: WebLogo of S5.1 is based on 4 DRs of which only the last shows a marked nucleotide variability. This single variability weighs numerically on the low number of detected DRs and accounts for the low level of conserved DRs in S5.1. Figure 3 has now been removed from the main text and presented in supplementary data as figures S1 (subtype I-E) and S2 (subtype I-F). In these figures we also report (as requested by the reviewer 1) the WebLogo of DRs identified in the complete genomes of S. marcescens available in the NCBI database.

Comment 4: In conjunction with the lack of obvious phage-derived spacers (Table 2) and with its length, how sure are the authors that this is a real array? Any additional evidence would be welcome.

Reply: a list of plasmids the spacers of S5.1 match is reported in File S2. A description of matched plasmids isolated in S. marcescens is now reported in the section results. According to what was published by Zhang and Ye (DOI 10.1186/s12859-017-1512-4) on the criteria for distinguishing real-CRISPRs from false-CRISPRs, a CRISPR array associated to cas genes can be considered real.

Comment 4:  The tree in Figure 4 is shown as rooted, with S. marcescens RSC-14 as an outgroup. Was there any particular reason to place the root there? If the tree is actually unrooted, it should rather be depicted as such.

Reply: the Figure 4 has been renamed as Figure 2. The tree has now been converted from a rectangle cladogram to a radial tree.

Round 2

Reviewer 1 Report

The authors changes have improved the manuscript sufficiently for publication.